# Workplace impact on employees: A Lifelines Corona Research Initiative on the return to work

Mark P. Mobach[1,2]*, Lifelines Corona Research Initiative[3¶]

1 Hanze University of Applied Sciences Groningen, Groningen, The Netherlands, 2 The Hague University of Applied Sciences, The Hague, The Netherlands, 3 Lifelines, University Medical Centre Groningen, University of Groningen, Groningen, The Netherlands

¶ Membership of the Lifelines Corona Research Initiative consortium is provided in the Acknowledgments.
* m.p.mobach@pl.hanze.nl

**Data Availability Statement:** The manuscript is based on data from the Lifelines Cohort Study, Study OV20_00070. Lifelines adheres to standards for data availability. Due to ethical restrictions

## Abstract

A large proportion of the global workforce migrated home during the COVID-19 pandemic and subsequent lockdowns. It remains unclear what the exact differences between home workers and non-home workers were, especially during the pandemic when a return to work was imminent. How were building, workplace, and related facilities associated with workers' perceptions and health? What are the lessons to be learned? Lifelines Corona Research Initiative was used to compare employees' workplaces and related concerns, facilities, work quality, and health in a complete case analysis (N = 12,776) when return to work was imminent. Mann-Whitney U, logistic regression, and Wilcoxon matched-pairs were used for analyses. Notwithstanding small differences, the results show that home workers had less favourable scores for concerns about and facilities of on-site buildings and workplaces upon return to work, but better scores for work quality and health than non-home workers. However, additional analyses also suggest that building, workplace, and related facilities may have had the capacity to positively influence employees' affective responses and work quality, but not always their health.

## Introduction

In an unprecedented change to the workplace, a large proportion of the global workforce migrated home during the COVID-19 pandemic and subsequent lockdowns. For most employees, this was a radical and an unexpected prolonged change to their work situation. The exact differences between employees working from home and those not working from home, especially during the pandemic when a return to work was imminent, remain unclear. What was the impact of different workplaces on employees? What are the lessons to be learned? In other words, did employees working from home report different concerns, facilities, work quality, and health shortly before returning to work during the COVID-19 pandemic than did employees not working from home?

imposed by the Lifelines Scientific Board and the Medical Ethical Committee of the University Medical Center Groningen related to protecting patient privacy, the data are not publicly available. The data catalogue of Lifelines is publicly accessible on https://www.lifelines.nl/researcher/data-and-biobank/$6102/$6104. All international researchers can obtain data at the Lifelines research office (research@lifelines.nl), for which a fee is required. The Lifelines system allows access for reproducibility of the study results.

**Funding:** The Lifelines Biobank initiative has been made possible by subsidy from the Dutch Ministry of Health, Welfare and Sport, the Dutch Ministry of Economic Affairs, the University Medical Center Groningen (UMCG the Netherlands), University Groningen and the Provinces Groningen, Friesland, and Drenthe of Northern Netherlands. The funders had no role in study design, data collection and analysis, decision to publish, or preparation of the manuscript.

**Competing interests:** The authors have declared that no competing interests exist.

The buildings and workplaces of organizations influence employees in different sectors: for instance, in healthcare [1, 2], higher education [3, 4], offices [5, 6], and cleaning industries [7, 8]. The management of buildings and workplaces is the responsibility of facility management, a discipline which focuses on the operations phase of a building [9] and examines how spaces and related services interact with the organization and people [10]. Where to work and how to facilitate people best in doing their jobs has been investigated widely. Studies suggest that the workplace has the capacity to advance the health, wellbeing, and safety of employees [11] as well as their satisfaction and productivity [12]. Buildings, workplaces, and related facilities are generally assumed to be unable to change the basics of the primary processes of organizations, but they can most certainly facilitate, hinder, or frustrate these processes [13].

Before the pandemic, most people worked on-site. In 2019, 60% of employees were working entirely on-site [14]. During the pandemic, protective workplace measures were actively used by authorities to mitigate the spreading of disease [15]. If possible, employees had to work from home. However, more than half of all employees did not work from home, simply because they were unable to do so [16]. Workers with critical jobs, for instance, in healthcare, supermarkets, and cleaning industries, worked on-site. Also, workers in the service, transport, and logistics sectors, and in agriculture, worked relatively little from home [17]. In March 2020, teleworking became an overnight reality for countless workers, particularly those in high-income countries, as exemplified by a 61% decrease in US elevator traffic in March 2020 [18]. Teleworking numbers rose, ranging from 13% of Brazil's workforce to approximately 33% and 50% of the workforce in Europe and the US, respectively [19].

During the pandemic, reports of employee experiences varied widely, basically boiling down to those for and against working from home [20, 21]; the implications for the management of workplaces, facilities, and real estate continue to exist [10, 22]. Workers and management disagree on whether or not to remain working from home, or to what extent [23]. These differences of opinion are mitigated by the use of blended working, where employees blend on-site working with working from home and in other places [24].

The COVID-19 pandemic and subsequent lockdowns have forced many employees to work from home. Lessons learned from this unparalleled global workplace experiment remain scarce, especially in view of possible future virus outbreaks [25]. Knowledge of workers' spatially-related experiences and health is particularly scarce. It remains unclear what the exact differences between home workers and non-home workers were with respect to their concerns, facilities, work quality, and health during the pandemic when a return to work was imminent. Moreover, it is also unknown how the places where people work and to what extent they are facilitated and feel concerned has influenced their work quality and health.

The aim of this study was to report on differences in experiences and health between home workers and non-home workers just before a return to work during the COVID-19 pandemic. Statistically significant differences between the two groups were observed and are reported. A peculiar contribution of this study is the finding that building, workplace, and related facilities seem to have had the capacity to positively influence employees' affective responses and work quality, but not always their health.

Below, we first provide an underpinning of our expectations in this study with a concise overview of the relevant literature for each factor. As data were collected in the summer of 2020, the main focus was on workplace-related studies in that year. Next, the context and the design of the study, the hypotheses, and the measures and methods used are justified and described. Following this, the results are presented. As the data consisted of the responses of participants who filled in the questionnaire either once (July or September 2020) or twice (July and September 2020), single measures and repeated measures are analyzed and described separately. Finally, the results are discussed and conclusions are drawn.

## Working from home

The Organisation for Economic Co-operation and Development [26] reported that in Australia, France, and the UK, 47% of employees worked from home during lockdowns in 2020. Wigert [27] reported that the proportions of employees who worked exclusively from home before, during, and after the pandemic were 8%, 70%, and 39%, respectively, highlighting a working-from-home boom during the pandemic. Researchers in the Netherlands have also estimated the proportions of home workers during the COVID-19 pandemic and subsequent lockdowns in 2020. The findings were much lower, varying from 41% of the Dutch workforce [28] to 46% [16] and 49% [29].

A December 2020 survey showed that 72% of US workers preferred to work from home [30], although drawbacks reported by European workers highlight missing out on workplace qualities, decreased meaningfulness of the work, and concerns about inadequate work supplies [31, 32], and, in general, significant economic loss [33, 34]. Working from home requires, e.g., high-quality ICT, a well-equipped home office, and managerial trust [35–37].

In the current study, it was expected that employees who were working from home would have different experiences than employees who were not working from home.

## Workplace concerns

A July 2020 global survey initiated by the International Facility Management Association showed that 40% of facility managers were concerned about HVAC systems, and 31% about the interiors of their premises [38]. Rothe and Hanc [39] reported that 7% and 11% of on-site-only workers and home workers, respectively, had health and safety concerns about their workplace. Yet another study reported that 64% of workers were unwilling to go back to the workplace, for reasons including safety fears and loss of productivity; 56% of employees in the Netherlands were reluctant to return to work [40].

It was expected, therefore, that home workers would have greater concerns about their workplace upon return to work than employees who were not working from home and were used to it.

## Workplace facilities

During the pandemic, measures were introduced for on-site workers to contain infection risks [41] and to safeguard their health and wellbeing [42]. Behavioural rules were adopted: e.g., social distancing, disinfecting of hands and contact points, and ventilating with fresh air [43–45]. Social distancing was furthered using one-way traffic signage, spatial-numerical occupant limitation, and an increase of building access points [46]. Dividers, glass-partitions, and acrylic screens were placed where close proximity was unavoidable [47]. Face masks and protective suits were prescribed for personal protection: e.g., in the healthcare and cleaning industries [48]. Cleaning activities were intensified; liquids and other cleaning materials were supplied for personal hygiene and the self-cleaning of workstations, thus advancing the cleanliness of workers and workplaces, and mitigating the risk of disease transmission [49].

During lockdown, such measures were limited to critical jobs. When the reproduction numbers and prevalence of COVID-19 infections decreased and the related rules and regulations became more relaxed, similar measures emerged in the buildings and workplaces of organizations to which staff returned in the course of 2020. Measures were to be followed strictly, as non-compliance with rules and regulations could potentially lead to the closing down of buildings and/or businesses [50, 51].

In the summer of 2020, as home workers had got used to working in the relative safety of the home environment and to using facilities to contain infection risks themselves, it was

hypothesized that upon return to work, these home workers would rate their on-site workplace facilities lower than would employees who were not working from home.

## Work quality

Workers have developed a good sense of where they can work successfully [52]. Different studies suggest that working from home usually leads to increased work productivity [53–55]. A 2020 survey, held between June and October among US and German employees, showed an average of 14% perceived productivity increase when working from home [56]. Reported reasons for such increases are fewer interruptions by colleagues [57], increased well-being [35], reduced commuting time [55], improved control (working hours, organisation of work), and better work-life balance [58].

Even though these studies seem to justify positive expectations, it is still unclear to what extent working from home influences perceived work quality. However, as with productivity, it was expected that home workers would report better current work quality, compared with the quality before corona, than would employees who were not working from home.

## Health

Various studies have shown that the places where people work have significant effects on their health [59–62] and wellbeing [63–65]. In the pre-pandemic era, Bloom et al. [54] reported that home working led to fewer sick days. It remains unclear, however, if this decrease is a direct and positive consequence of working from home, or if employees choose to work when sick [66, 67]. During the pandemic, working from home was associated with reduced disease transmission [49], but also with a downside of decreased mental well-being [58, 68]. Moreover, it has been argued that working from home had comparatively limited effects in curbing COVID-19 spread [69, 70].

These cautions notwithstanding, it was expected that home workers would report lower sick leave and fewer COVID-19 infections or symptoms since the start of the pandemic than employees who were not working from home.

# Materials and methods

## Context of the study

On June 1 2020, the Dutch authorities liberalized the COVID-19 regulations, which then remained relatively stable until the second wave in October 2020 [71]. During the summer, organizations started to prepare for a returning workforce, mostly by trial and error, because knowledge of how to respond to the challenges in buildings and how to anticipate new outbreaks was still lacking.

In the Netherlands, the largest changes occurred within the groups of home workers. Comparing the last quarters of 2019 and 2020, Netherlands Environmental Assessment Agency [16] showed increases of 5% and 11% for home workers and exclusive home workers, respectively, indicating that more people worked from home and also that the numbers of those working exclusively from home increased gradually over 2020.

## Research design

In this study, the associations between workplace-related factors (workplace, concerns, facilities), work (quality), and health (sick leave, COVID-19) were investigated among 10,889 workers in the Netherlands (Fig 1).

**Population.** The study was conducted using data from the Lifelines Cohort Study. Lifelines is a multi-disciplinary prospective population-based cohort study examining in a unique

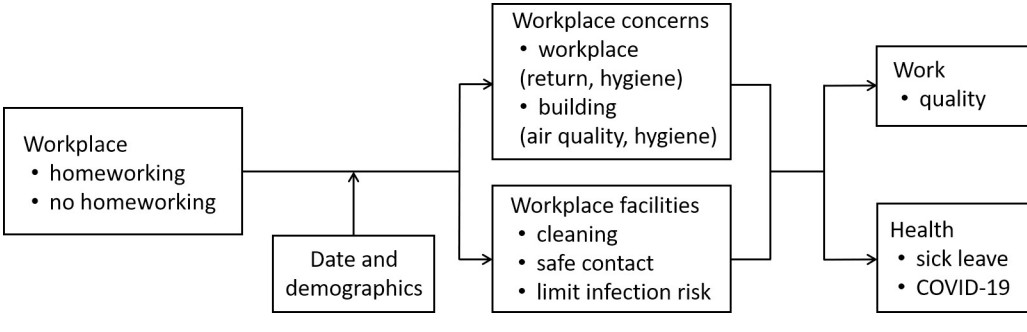

**Fig 1. Research model.**

three-generation design the health and health-related behaviours of 167,729 persons living in the north of the Netherlands. It employs a broad range of investigative procedures in assessing the biomedical, socio-demographic, behavioural, physical, and psychological factors which contribute to the health and disease of the general population, with a special focus on multi-morbidity and complex genetics. Scholtens et al. [72] describe the scientific rationale, study design, and survey methods of the Lifelines Cohort Study. Lifelines data were supplemented with additional questionnaire data collected by the Lifelines Corona Research Initiative, using the Dutch Lifelines COVID-19 cohort study (N = 17,749).

The Lifelines adult study population is broadly representative with respect to socioeconomic characteristics, lifestyle factors, the prevalence of chronic diseases, and general health in the north of the Netherlands. Before entering the study, written informed consent was obtained from all participants. The LifeLines Cohort Study was conducted according to the principles of the Declaration of Helsinki and in accordance with the research code of University Medical Center Groningen (UMCG). The LifeLines study was approved by the medical ethical committee of the UMCG, the Netherlands. For a comprehensive overview of the data collection, please visit the LifeLines catalogue at www.LifeLines.net.

**Procedure.** IBM SPSS Statistics Version 25.0.0.1 was used for statistical analyses. Respondents with missing data were excluded from the analysis, resulting in a complete case analysis with a sample of 12,776 responses collected in July and September 2020; 9,002 respondents filled in the questionnaire once; 1,887 respondents filled it in twice. Participants' Gender and Age were taken from the Lifelines baseline screening. Workplace, Workplace Concerns, Workplace Facilities, Work Quality, Sick Leave, and COVID-19 were extracted from the COVID-19 questionnaires of Lifelines. All items for Workplace Concerns and Workplace Facilities were clearly labelled for the respondents under the category 'Back to work'.

**Hypothesis.** It was hypothesized that home workers would report significantly less favourable scores for Workplace Concerns and Workplace Facilities when faced with the prospect of returning to the workplace, but better scores for Work Quality and Health in their current situation than non-home workers. At first sight, this hypothesis may seem counterintuitive; it seems more logical to expect that home workers would also report fewer concerns and better facilities than non-home workers during the pandemic. However, it was hypothesized that these workers might have a very different perspective when focusing on the building to which they had to return, especially immediately before the return of all staff.

## Measures

**Date and demographics.** Date comprised month of the assessment (July 2020; September 2020); Demographics consisted of the gender and age of respondents.

**Workplace.**    Workplace was measured using one item on a dichotomous scale (1 'I work from home'; 2 'I do not work from home').

**Workplace concerns.**    A multi-item scale was composed to measure Workplace Concerns. This scale comprised four items: 'I am worried about returning to my workplace', 'I am worried about the hygiene in the building where I work', 'I am worried about the air quality in the building where I work', and 'I am worried about the hygiene at my workplace'. These items were answered using a 7-point Likert scale (1 completely agree; 2 agree; 3 somewhat agree; 4 neutral; 5 somewhat disagree; 6 disagree; 7 completely disagree). The responses were recoded (reversed) so that higher scores reflected more agreement. The sum score of all items was also used for analyses (min = 4, max = 28). Cronbach's alpha was .86, which is acceptable [73, 74].

**Workplace facilities.**    Workplace Facilities was also measured using a multi-item scale, in this case with three items: 'I have sufficient supplies to clean my work things', 'I can safely have contact with people at my work', and 'I can limit the risk of infection myself in the building where I work'. Scales and recoding were the same as for Workplace Concerns. The sum score of all items was used for analyses (min = 3, max = 21). Cronbach's alpha was .76, which is also acceptable [73, 74].

**Work quality.**    For Work Quality, a single-item scale was applied, using a 5-point Likert scale: 'Is the quality of your work worse, the same, or better than before the corona crisis?' (1 much worse; 2 worse; 3 the same; 4 better; 5 much better).

**Health.**    Sick Leave was also measured using one item, in this case on a dichotomous scale: 'Have you called in sick or taken leave of absence since the start of the corona crisis (mid-March)?' (1 yes; 2 no). Finally, the reason for sick leave was asked: 'Why did you call in sick or take leave of absence'? (1 due to corona (infection or symptoms); 2 other reasons).

Workplace, Work Quality, Sick Leave, and COVID-19 were measured using a single-item scale; this was considered acceptable, because the constructs were regarded as sufficiently clear, narrow, and unambiguous [70, 75].

**Method justification.**    A Kolmogorov-Smirnov test, descriptive statistics, Mann-Whitney U test, Chi-Square test, Belsley's collinearity diagnostics, logistic regression, Hayes' process procedure, Wilcoxon matched pairs signed-ranks analysis, McNemar's test, and Sign test were used consecutively in the analyses. Justification for these choices is given below.

The Kolmogorov-Smirnov test of normality was used to test whether the data were normally distributed, allowing for parametric methods. Because this was not the case, the main analyses consisted of descriptive statistics, non-parametric methods, and logistic regression models.

Non-parametric methods were used because these methods do not require a normal distribution. The data were split into two files: single measures (responses from July or September) and repeated measures (responses from July and September). These data files were analyzed using different methods. Both were combined with descriptive statistics to explore and describe the different sample characteristics.

Mann-Whitney U was used for single measures: to test for differences between home workers and non-home workers. This non-parametric method was applied because it allows for a comparison of differences between two independent groups (i.e., the responses of different participants who were categorized as home and non-home workers) with ordinal (i.e., Workplace Concerns, Workplace Facilities, Work Quality) or continuous (i.e., Age) dependent variables. Because categorical dependent variables (i.e., Date, Gender, Sick Leave, COVID-19) require a different approach, a Chi-Square test was also used to confirm the findings.

Logistic regression models were applied to the same sample to refine our understanding of how the three dependent variables Work Quality, Sick Leave, and COVID-19 were influenced by our independent variables. Belsley's collinearity diagnostics was used to test the predictor

variables a priori and verify that there was no violation of the assumption of no multicollinearity. This was confirmed, so logistic regression was possible. These models were used because they do not assume normality and they function well in predicting the probability of an event taking place in one of the two categories of a dichotomous dependent variable (i.e., Sick Leave, COVID-19) or in more than two categories of an ordinal dependent variable (i.e., Work Quality), given more independent variables. Hence, binary logistical regression and ordinal logistical regression models, respectively, were used. Interactions between the main independent variables of interest (i.e., Workplace, Workplace Concerns, Workplace Facilities) and the dependent variables (Fig 1) were analyzed to determine if any of the main effects was influenced by a third variable. In the event that this occurred, Hayes' process analysis was used to scrutinize the observed relationships and to determine what their exact meaning was.

Wilcoxon matched pairs was used for repeated measures: to test for differences between July and September. This non-parametric method was used because it allows for a comparison of differences between two dependent groups (i.e., the responses of the same participants from July and September, who were also categorized as home and non-home workers) with ordinal dependent variables (i.e., Workplace Concerns, Workplace Facilities, Work Quality). Because categorical dependent variables (i.e., Sick Leave, COVID-19) require a different approach, McNemar's test was used to confirm the findings. In the case of a cell frequency equal to zero, McNemar's test could not be performed and was replaced with Sign test.

## Statistical analyses

Single and repeated measures were treated separately for comparative analyses [76]. An important reason for splitting the data file was that different methods can be employed for single and repeated measures. For a logistic regression, for instance, independent observations are required, which means that they should not come from repeated or paired data; whereas repeated measures allow for within-person comparisons, e.g., using Wilcoxon matched pairs.

A Kolmogorov-Smirnov test of normality was conducted to determine whether items were normally distributed. It showed that the null hypothesis had to be rejected for all items. Data were not normally distributed, neither for the single measures with 9,002 respondents ($p < .01$) nor for the repeated measures with 1,887 respondents ($p < .01$). Consequently, a Mann-Whitney U test ($p < .01$) was performed to compare the responses of home workers and non-home workers to the prospect of returning to work with regard to Date and Demographics, Workplace Concerns, Workplace Facilities, Work Quality, Sick Leave, and COVID-19. The $z$ values were used to calculate effect sizes and, as proposed by Cohen [77], $r$ was calculated. We interpreted $r$ as follows: .5 indicates a large effect, .3 a medium effect, and .1 a small effect [78]; reversed for negative scores.

A Wilcoxon matched pairs signed-ranks analysis ($p < .01$) was used to compare responses over time. Individuals' responses were matched into 1,887 pairs. This classification allowed a comparison of the same individuals' responses regarding Workplace, Workplace Concerns, Workplace Facilities, Work Quality, Sick Leave, and COVID-19 in July and in September 2020. Effect sizes were calculated and interpreted as for the Mann-Whitney U test.

Logistical regression models were constructed to analyze the hypothesized relationships, as these models do not assume normality. The predictor variables were tested a priori to verify that there was no violation of the assumption of no multicollinearity, following the criteria of Belsley et al. [79]: Tolerance > .1, Variance Inflation Factor (VIF) < 10, Condition Index < 30. The tests showed that the data met the assumption of collinearity, indicating that multicollinearity of our independent variables was not a concern. The respective scores were Workplace (Tolerance = .952, VIF = 1.051, Condition Index = 5.304), Date (Tolerance = .979,

VIF = 1.021, Condition Index = 8.147), Gender (Tolerance = .961, VIF = 1.041, Condition Index = 9.447), Age (Tolerance = .986, VIF = 1.014, Condition Index = 10.463), Workplace Concerns (Tolerance = .779, VIF = 1.284, Condition Index = 28.190), and Workplace Facilities (Tolerance = .810, VIF = 1.235, Condition Index = 13.028). The Box-Tidwell test was used to check for linearity between the continuous predictor Age and the logit. The result was not statistically significant ($p$ = .897), implying that this independent variable was linearly related to the logit of the outcome variables and that the assumption was satisfied.

Ordinal logistical regression was used to analyze Work Quality as a function of Workplace, Date, Demographics, Workplace Concerns, and Workplace Facilities. Binary logistical regression was used to analyze Sick Leave and COVID-19 as a function of Workplace, Date, Demographics, Workplace Concerns, and Workplace Facilities. In line with our research model (see Fig 1), two-way interactions and the three-way interaction were also included for Workplace, Workplace Concerns, and Workplace Facilities. Finally, Hayes' [80] process procedure was used to further analyze the statistically significant interactions between Workplace, Workplace Concerns, and Workplace Facilities; Models 1 and 3 were used for the two-way interactions and the three-way interaction, respectively: 5000 bootstrap samples were performed.

## Results

### Single measures N = 9,002

The Mann-Whitney U test showed that almost all differences between the sampling distributions of home workers and non-home workers were statistically significant ($p$ < .001), but also that many of these differences were rather small (Table 1). Statistically significant differences were only absent for Age and the item 'I can safely have contact with people at my work'. Consequently, a large variety of statistically significant differences was observed.

**Baseline.** With respect to the workplace, fewer workers were working from home (N = 2,526, 28.06%) than not working from home (N = 6,476, 71.94%) in the sample.

**Date and demographics.** There were more women in the sample (N = 5,452, 60.56%). There were significantly more home workers in July (N = 1,675, 66.31%) than in September (N = 851, 33.69%, $z$ = -13.162, $p$ < .001). Significantly fewer female workers were working from home (female; N = 1,393, 55.15%) than not working from home (female; N = 4,059, 62.68%, $z$ = -6.569, $p$ < .001). Age did not differ significantly between home workers (M = 51.05, SD = 8.84) and non-home workers (M = 50.67, SD = 9.09, $z$ = -1.599, $p$ = .110). A Chi-Square test of the categorical variables confirmed the statistical significance of the above findings (Date ($\chi^2(1)$ = 173.257, $p$ < .001); Gender ($\chi^2(1)$ = 43.153, $p$ < .001)).

**Workplace concerns.** As hypothesized, Workplace Concerns regarding the built environment at work were significantly higher among home workers (M = 13.77, SD = 6.70) than among non-home workers (M = 11.74, SD = 6.37, $z$ = -13.274, $p$ < .001). Home workers were more concerned about the return to their workplace (homeworkers: M = 3.33, SD = 1.90 versus non-home workers: M = 2.65, SD = 1.78, $z$ = -16.292, $p$ < .001), the hygiene of the building where they worked (home workers: M = 3.24, SD = 1.90 vs non-home workers: M = 2.91, SD = 1.84, $z$ = -7.870, $p$ < .001), the air quality in their building (home workers: M = 3.93, SD = 2.01 vs non-home workers: M = 3.29, SD = 1.97, $z$ = -13.604, $p$ < .001), and the hygiene at their workplace (home workers: M = 3.27, SD = 1.91 vs non-home workers: M = 2.89, SD = 1.82, $z$ = -9.024, $p$ < .001).

**Workplace facilities.** Also as hypothesized, the scores for Workplace Facilities within the built environment at work were significantly lower for home workers (M = 14.83; SD = 3.94) than for non-home workers (M = 15.12, SD = 4.15, $z$ = -4.004, $p$ < .001). Home workers reported having fewer sufficient supplies to clean their work things (homeworkers: M = 5.16,

**Table 1. Descriptive statistics and Mann-Whitney U test for statistical significance of the differences in Workplace with regard to sample characteristics, Workplace Facilities, Workplace Concerns, Work Quality, Sick Leave, and COVID-19; single measures, N = 9,002 (70.46%).**

| Variables | Workplace; N (%) | | $z$ | $p$ | $r$ |
|---|---|---|---|---|---|
| | Home; 2526 (28.06) | Non-home; 6476 (71.94) | | | |
| Date (month, year); N (%) | | | -13.162 | < .001 [a] | -.139 |
| July 2020 | 1675 (33.67) | 3300 (66.33) | | | |
| September 2020 | 851 (21.13) | 3176 (78.87) | | | |
| Gender; N (%) | | | -6.569 | < .001 [a] | -.069 |
| Male | 1133 (44.85) | 2417 (37.32) | | | |
| Female | 1393 (55.15) | 4059 (62.68) | | | |
| Age (years); M (SD) | 51.05 (8.84) | 50.67 (9.09) | -1.599 | .110 | -.017 |
| Workplace Concerns; M (SD) | 13.77 (6.70) | 11.74 (6.37) | -13.274 | < .001 | -.139 |
| I am worried about returning to my workplace; M (SD) | 3.33 (1.90) | 2.65 (1.78) | -16.292 | < .001 | -.172 |
| I am worried about the hygiene in the building where I work; M (SD) | 3.24 (1.90) | 2.91 (1.84) | -7.870 | < .001 | -.083 |
| I am worried about the air quality in the building where I work; M (SD) | 3.93 (2.01) | 3.29 (1.97) | -13.604 | < .001 | -.143 |
| Workplace Facilities; M (SD) | 14.83 (3.94) | 15.12 (4.15) | -4.004 | < .001 | -.042 |
| I have sufficient supplies to clean my work things; M (SD) | 5.16 (1.60) | 5.47 (1.57) | -10.294 | < .001 | -.108 |
| I can safely have contact with people at my work; M (SD) | 5.02 (1.51) | 4.89 (1.73) | -1.057 | .290 | -.011 |
| I can limit the risk of infection myself in the building where I work; M (SD) | 4.66 (1.69) | 4.76 (1.74) | -3.222 | .001 | -.034 |
| I am worried about the hygiene at my workplace; M (SD) | 3.27 (1.91) | 2.89 (1.82) | -9.024 | < .001 | -.095 |
| Work Quality; M (SD) | 3.06 (0.49) | 2.96 (0.33) | -9.931 | < .001 | -.105 |
| Sick Leave; N (%) | | | -4.491 | < .001 [a] | -.047 |
| Yes | 138 (5.46) | 533 (8.23) | | | |
| No | 2388 (94.54) | 5943 (91.77) | | | |
| COVID-19; N (%) | | | -6.045 | < .001 [a] | -.064 |
| Yes | 12 (8.70) | 187 (35.08) | | | |
| No | 126 (91.30) | 346 (64.92) | | | |

[a] $p < .001$ for Chi-Square Test.

$SD$ = 1.60 versus non-home workers: $M$ = 5.47, $SD$ = 1.57, $z$ = -10.294, $p < .001$) and fewer facilities to limit the risk of infection in the building where they worked (homeworkers: $M$ = 4.66, $SD$ = 1.69 vs non-home workers: $M$ = 4.76, $SD$ = 1.74, $z$ = -3.222, $p < .001$). Only the item 'I can safely have contact with people at my work' showed no statistically significant differences (home workers: $M$ = 5.02, $SD$ = 1.51 vs non-home workers: $M$ = 4.89, $SD$ = 1.73, $z$ = -1.057, $p$ = .290).

**Work quality.** As expected, reported Work Quality was slightly higher among home workers ($M$ = 3.06, $SD$ = 0.49) than among non-home workers ($M$ = 2.96, $SD$ = 0.33, $z$ = -9.931, $p < .001$); this difference was statistically significant. Remember, however, that a value of 3 indicated that current quality of the work was the same as before the corona crisis.

**Health.** Also as hypothesized, both Sick Leave and COVID-19 showed statistically significant differences between home workers and non-home workers. Reported sick leave was well over 2.5% lower among home workers (N = 138, 5.46%) than among non-home workers (N = 533, 8.23%, $z$ = -4.491, $p < .001$). Sick leave due to COVID-19 was also significantly lower among home workers (N = 12, 8.70%) than among non-home workers (N = 187, 35.08%, $z$ = -6.045, $p < .001$): more than 25% lower. A Chi-Square test of these two categorical variables confirmed the statistical significance of the above findings (Sick Leave ($\chi^2(1)$ = 20.172, $p < .001$); COVID-19 ($\chi^2(1)$ = 36.591, $p < .001$)).

Apart from Age and the item 'I can safely have contact with people at my work', all reported differences were statistically significant. An additional analysis showed that when all repeated

measures from the data analysis were included (N = 12,776), the statistical significance of all of these findings remained the same ($p < .01$). However, even though most differences between home workers and non-home workers were statistically significant, many of the observed differences were rather small. For all items, $r$ was between -.2 and 0, indicating a small effect size [77, 78].

Ordinal and binary logistical regression models were constructed to analyze the hypothesized relationships. Ordinal logistical regression was used to analyze Work Quality as a function of Workplace, Date, Demographics, Workplace Concerns, and Workplace Facilities. Binary logistical regression was used to analyze Sick Leave and COVID-19 as a function of Workplace, Date, Demographics, Workplace Concerns, and Workplace Facilities. In line with our research model (see Fig 1), main effects, two-way interactions, and a three-way interaction were included for all regression analyses. The three models showed a significant improvement in fit when the predictors were used ($p < .0001$, Omnibus Test). The models explained 3.62% of the variance in Work Quality, 4.10% in Sick Leave, and 16.05% in COVID-19 (Nagelkerke $R^2$), respectively.

The $p$ values of the ordinal logistical regression model indicate that all main effects and interaction effects were significant ($p < .01$), except for the main effect Age ($p = .526$). This means that the model identified Workplace, Date, Gender, Workplace Concerns, and Workplace Facilities as explanatory variables for Work Quality. Table 2 shows that increased scores

**Table 2. Results of logistic regression analyses on Work Quality, Sick Leave, and COVID-19, single measures, N = 9,002.**

| Source of variation | Work Quality[a] | | | | Sick Leave[b] | | | | COVID-19[b] | | | |
|---|---|---|---|---|---|---|---|---|---|---|---|---|
| | B (SE) | Wald $\chi^2$ (df) | p | OR (CI: 95%) | B (SE) | Wald $\chi^2$ (df) | p | OR (CI: 95%) | B (SE) | Wald $\chi^2$ (df) | p | OR (CI: 95%) |
| (intercept) | | | | | -4.489 (1.956) | 5.265 (1) | .022 | .011 (.000, .520) | 1.941 (9.382) | .043 (1) | .836 | 6.965 (.000, 673845333.000) |
| Workplace; home | 1.835 (.620) | 8.758 (1) | .003 | 6.264 (1.858, 21.115) | .855 (1.023) | .699 (1) | .403 | 2.352 (.317, 17.452) | -1.555 (4.743) | .107 (1) | .743 | .211 (.000, 2301.322) |
| Date; Sept | -.195 (.062) | 9.717 (1) | .002 | .823 (.728, .930) | .568 (.083) | 47.037 (1) | < .001 | 1.765 (1.501, 2.077) | .950 (.197) | 23.215 (1) | < .001 | 2.586 (1.757, 3.807) |
| Gender; female | -.157 (.064) | 5.951 (1) | .015 | .855 (.754, .970) | .208 (.088) | 5.629 (1) | .018 | 1.232. (1.037, 1.463) | .108 (.202) | .285 (1) | .593 | 1.114 (.749, 1.656) |
| Age | .002 (.003) | .402 (1) | .526 | 1.002 (.995, 1.009) | -.028 (.004) | 43.061 (1) | < .001 | .972 (.964, .981) | -.026 (.009) | 7.685 (1) | .006 | .974 (.957, .992) |
| Workplace Concerns | .326 (.063) | 27.100 (1) | < .001 | 1.385 (1.225, 1.556) | .047 (.101) | .220 (1) | .639 | 1.048 (.860, 1.278) | -.390 (.496) | .616 (1) | .432 | .677 (.256, 1.792) |
| Workplace Facilities | .202 (.068) | 8.959 (1) | .003 | 1.224 (1.072, 1.397) | .077 (.113) | .458 (1) | .498 | 1.080 (.865, 1.347) | -.443 (.567) | .611 (1) | .434 | .642 (.211, 1.951) |
| Workplace x Concerns | -.193 (.035) | 29.961 (1) | < .001 | .824 (.769, .883) | .003 (.054) | .003(1) | .956 | 1.003 (.902, 1.116) | .193 (.252) | .584 (1) | .445 | 1.212 (.740, 1.987) |
| Workplace x Facilities | -.102 (.037) | 7.594 (1) | .006 | .903 (.840, .971) | -.033 (.060) | .300 (1) | .584 | .968 (.860, 1.088) | .218 (.287) | .575 (1) | .448 | 1.243 (.708, 2.183) |
| Concerns x Facilities | -.015 (.004) | 12.535 (1) | < .001 | .985 (.977, .993) | -.001 (.007) | .049 (1) | .825 | .999 (.986, 1.011) | .029 (.031) | .847 (1) | .357 | 1.029 (.968, 1.095) |
| Workplace x Concern x Facilities | .008 (.002) | 12.346 (1) | < .001 | 1.008 (1.004, 1.013) | .0001 (.004) | .002 (1) | .968 | 1.000 (.993, 1.007) | -.014 (.016) | .762 (1) | .383 | .986 (.956, 1.017) |

[a] Ordinal logistic regression.

[b] Binary logistic regression.

*Note*. B: Co-efficient for the constant; SE: standard error around the co-efficient for the constant; Wald $\chi^2$: Wald chi square statistics; df: degree of freedom for Wald chi square statistics; OR: odds ratio; CI: 95% confidence interval for the odds ratio with its upper and lower limits.

in Work Quality corresponded with higher scores on home working (odds ratio = 6.264), Workplace Concerns (odds ratio = 1.385), and Workplace Facilities (odds ratio = 1.224). The highest odds ratio for Work Quality (OR = 6.264; 95% CI = 1.858, 21.115) suggests that the odds of better work quality were 6.26 times higher among home workers than non-home workers.

In contrast, the *p* values of the binary logistical regression model of Sick Leave indicate that all main effects and interaction effects were *not* significant ($p < .01$), except for the main effects Date (*B* = .568, $p < .001$), Gender (*B* = .208, $p = .018$), and Age (*B* = -.028, $p < .001$). Similarly, the *p* values of the binary logistical regression model of COVID-19 indicate that only the main effect Date (*B* = .950, p < .001) and Age (*B* = -.026, p < .01) were statistically significant. These latter two models only identified Date, Gender, and Age as explanatory variables for Sick Leave, and Date and Age for the prevalence of COVID-19. Removal of outliers (N = 12) had no effect on the statistical significance of the results ($p < .01$).

Next, statistically significant interaction effects were further analyzed using Hayes' process analysis [80]. Three two-way interactions (Model 1) and one three-way interaction (Model 3) were analysed; 5,000 bootstrap samples were performed. The results showed, firstly, for home workers, that an increase in reported Workplace Concerns resulted in a small but statistically significant increase in Work Quality (*b* = .005, $p < .001$); for non-home workers, an increase in reported Workplace Concerns resulted in a small significant decrease in Work Quality (*b* = -.005, $p < .001$). Secondly, for non-home workers, an increase in reported Workplace Facilities yielded a small increase in Work Quality (*b* = .005, $p < .001$); for home workers, these effects were not significant (*b* = -.004, $p = .168$). Thirdly, for home workers scoring low for Workplace Concerns, an increase in reported Workplace Facilities resulted in a small increase in Work Quality (*b* = .004, $p = .005$); the results were not statistically significant for home workers scoring average (*b* = .002, $p = .093$) or high (*b* = .0003, $p = .897$) for Workplace Concerns. Fourthly, home workers scoring moderate or high for Workplace Concerns, regardless of their scores for Workplace Facilities, showed a small decrease in Work Quality ($-.100 < b < -.074$, $p < .01$). Moreover, home workers with low scores for Workplace Concerns and high scores for Workplace Facilities, also showed a small and statistically significant decrease in Work Quality (*b* = -.042, $p = .024$); when these workers scored low or medium for Workplace Facilities, these effects were not significant (*b* = .023, $p = .455$ and *b* = -.009, $p = .591$, respectively).

## Repeated measures N = 1,887

Wilcoxon matched-pairs signed-ranks analysis of the responses from July and September showed that most differences were not statistically significant (Table 3). Only five statistically significant differences were found. Over time, Workplace Concerns increased significantly among non-home workers (July M = 11.70, SD = 6.33; September M = 12.19, SD = 6.33; *z* = -3.653, $p < .001$, *r* = -.076), as did Sick Leave (July N = 56, 4.82%; September N = 110, 9.46%; *z* = -4.597, $p < .001$, *r* = -.095) and COVID-19 (July N = 11, .95%; September N = 42, 3.61%; *z* = -3.287, $p = .001$, *r* = -.068). Moreover, among workers who worked from home in July and switched to non-home working in September, Sick Leave also increased significantly (July N = 2, 1.48%; September N = 9, 6.67%; *z* = -2.333, $p = .020$, *r* = -.142), as did COVID-19 (July N = 0, 0%; September N = 1, 74%; *z* = -2.181, $p = .029$, *r* = -.133). Again, for all statistically significant items, *r* was between -.2 and 0, indicating a small effect size [77, 78]. Other statistical tests confirmed these outcomes. McNemar's test confirmed statistical significance for Sick Leave ($p < .001$) and COVID-19 ($p < .001$) among persistent non-home workers. Also, McNemar's test and Sign test confirmed statistical significance for Sick Leave ($p < .05$) and COVID-19 ($p < .05$), respectively, among home workers (July) who became non-home workers (September).

**Table 3. Descriptive statistics and Wilcoxon matched-pairs signed-ranks analysis for statistical significance of the differences between July and September within subjects with regard to Workplace Facilities, Workplace Concerns, Work Quality, Sick Leave, and COVID-19 for differences in Workplace; repeated measures, N = 1,887 (29.54%) matched pairs.**

| Variable | Workplace, n (%) | | | | | | | | | | | | | | | | | | | |
|---|---|---|---|---|---|---|---|---|---|---|---|---|---|---|---|---|---|---|---|---|
| | Home, n = 565 (29.94) | | | | | Non-home, n = 1163 (61.63) | | | | | Home (July), non-home (Sept), n = 135 (7.16) | | | | | Non-home (July), home (Sept), n = 24 (1.27) | | | | |
| | July | Sept | z | p | r | July | Sept | z | p | r | July | Sept | z | p | r | July | Sept | z | p | r |
| Workplace Concerns; M (SD) | 13.90 (6.77) | 13.83 (6.59) | -.535 | .593 | -.016 | 11.70 (6.33) | 12.19 (6.33) | -3.653 | < .001 | -.076 | 13.67 (6.81) | 13.28 (7.06) | -1.203 | .229 | -.073 | 13.71 (6.80) | 13.96 (6.62) | -.105 | .917 | -.015 |
| Workplace Facilities; M (SD) | 14.61 (4.06) | 14.94 (3.79) | -1.061 | .289 | -.032 | 15.22 (4.16) | 15.16 (3.96) | -.759 | .448 | -.016 | 15.18 (3.94) | 14.83 (4.60) | -.392 | .695 | -.024 | 15.00 (4.04) | 15.54 (3.85) | -.943 | .346 | -.136 |
| Work Quality; M (SD) | 3.09 (.46) | 3.07 (.47) | -1.173 | .241 | -.035 | 2.96 (.40) | 2.96 (.33) | -.200 | .841 | -.004 | 2.97 (.55) | 2.93 (.47) | -.955 | .339 | -.058 | 3.13 (.54) | 2.96 (.36) | -1.414 | .157 | -.204 |
| Sick Leave; N (%) | | | -1.333 | .182 | -.040 | | | -4.597 | < .001[a] | -.095 | | | -2.333 | .020[b] | -.142 | | | -1.633 | .102 | -.236 |
| Yes | 19 (3.36) | 27 (4.78) | | | | 56 (4.82) | 110 (9.46) | | | | 2 (1.48) | 9 (6.67) | | | | 1 (4.17) | 5 (20.83) | | | |
| No | 546 (96.64) | 538 (95.22) | | | | 1107 (95.18) | 1053 (90.54) | | | | 133 (98.52) | 126 (93.33) | | | | 23 (95.83) | 19 (79.17) | | | |
| COVID-19; N (%) | | | -1.055 | .291 | -.031 | | | -3.287 | .001[a] | -.068 | | | -2.181 | .029[c] | -.133 | | | -1.444 | .149 | -.208 |
| Yes | 0 (0) | 2 (.35) | | | | 11 (.95) | 42 (3.61) | | | | 0 (0) | 1 (.74) | | | | 0 (0) | 1 (4.17) | | | |
| No | 19 (3.36) | 25 (4.43) | | | | 45 (3.87) | 68 (5.85) | | | | 2 (1.48) | 8 (5.93) | | | | 1 (4.17) | 4 (16.66) | | | |

[a] $p < .001$ for McNemar's test.
[b] $p < .05$ for McNemar's test.
[c] $p < .05$ for Sign test.

## Discussion

The aim of this study was to investigate whether employees working from home had less favourable scores for concerns and facilities upon return to work, but better scores for work quality and health, than non-home workers. This was confirmed in the analyses; the findings were consistent with the hypotheses. However, detailed analyses also showed nuanced differences.

As hypothesized, all of our modelled factors (Fig 1) except age explained the quality of work of employees. In contrast, only date, gender, and age explained sick leave, whereas only date and age explained COVID-19. This implies that the place of work, concerns, and facilities did matter for work quality, but not for health. This may be explained by a gradual increase in the prevalence of COVID-19 between July and September 2020 [71, 81], people spending longer in isolation and reporting mental health issues [68], women being more likely to call in on sick leave [82], and older people being more likely to be ill for prolonged periods of time [82] and more likely to become infected with COVID-19 [83].

Interaction effects showed that home workers reported better work quality if they were more concerned about the on-site workplace. In this situation, work quality was lower for non-home workers. This may be explained simply by the fact that the latter actually worked in less favourable environments [49]; the same applies to non-home workers with better work-place facilities reporting better work quality. Home workers with few concerns and better facilities reported better work quality; however, home workers' work quality decreased when their concerns were moderate or high. In this context, fear of contamination with COVID-19 may have negatively affected employee's work performance [84]. It remains unclear, however, why home workers who scored low for on-site concerns and high for on-site facilities showed a decrease in current work quality.

Over time, workplace concerns, sick leave, and COVID-19 increased among non-home workers. As workers switched from home work to non-home working, sick leave and COVID-19 increased. This may reflect the higher risk of disease transmission and infection at on-site work [49].

### Strengths and limitations

This study has some strengths and limitations. One strength is that we had a large sample of quality data collected from a general working population of 10,889 workers; this strengthened the statistical power and generalisability of the findings. Second, to our knowledge, this is the first study in which the associations between workplace-related factors (i.e., workplace, work-place concerns, workplace facilities) and work quality, sick leave, and COVID-19 were investigated in such detail. Third, the sub-sample of 1,887 workers allowed for longitudinal comparisons between the workers' responses from July and September 2020. Fourth, the findings appear to be robust, as different sensitivity analyses did not change the findings. The study also has limitations: first, the findings relied on self-reports of respondents, which may have deviated from the data and findings of, e.g., directly observed, actual differences between these workers in the topics under investigation in organizations, the built environment, and at their workplaces [85]. Second, the return to work may have been implemented differently, at different paces, and with different (facility) management styles and quality in the different organizations where these people worked, possibly influencing their perceptions, ratings, and scores. Third, remember that the COVID-19 pandemic led to a special situation for the work-force, with specific rules and regulations for management, workers, facilities, and organizations. This may limit the applicability of these findings beyond the pandemic. All these effects may have influenced our results. Finally, most of the observed differences between employees

who did not work from home and home workers were statistically significant, but also relatively small. Consequently, we recommend caution in using the associations between workplace-related factors, work quality, and health reported in this context in the routine practices of organizations and facility departments in the Netherlands, not to mention other countries.

## Conclusions

Notwithstanding small differences, in comparison with employees who did not work from home, home workers were more concerned about their on-site workplace and reported lower quality of on-site facilities, better work quality than before COVID-19, and less sick leave and COVID-19 (infection or symptoms). Additional regression analyses confirmed that work quality can be explained by the factors in our model: employees' workplace, their workplace concerns, and the quality of the on-site facilities. However, health was not explained by our main factors. In contrast, our analyses of matched pairs showed that the concerns and health of non-home workers increased significantly over time, as did the health of home workers that switched to non-homeworking over time.

The results suggest that the built environment, the workplace, and related facilities were key to positively influencing the concerns and the quality of work, but not always the health, of employees during the pandemic. Caution is recommended when relating these results to the routine practices of organizations and facility departments, because the reported perceptions may have deviated from the actual properties of the work environment and workers' actual behaviours and health. Moreover, the COVID-19 pandemic led to a special situation for the workforce, with specific rules and regulations for management, workers, facilities, and organizations. Despite these cautions, the results imply that organizations should take into account the quality of the built environment, the workplace, and related facilities as these did positively influence employees' concerns and work, but not always their health, during the pandemic. Future research should reveal if the built environment may have similar influences on actual work-related behaviours and the health of employees in real-life settings and beyond the pandemic.

## Acknowledgments

The first author wishes to acknowledge the services of the Lifelines Cohort Study, the contributing research centres delivering data to Lifelines, and all study participants. Moreover, he thanks the Research Centre for Built Environment NoorderRuimte and Centre of Expertise Healthy Ageing of Hanze University of Applied Sciences, Groningen, the Netherlands for endorsing this research. Finally, the first author is grateful to the Lifelines Data Management Team for their valuable support and to the Lifelines Corona Research Initiative for their preparedness to include questions related to facility management and make these available for analysis.

## Consortium section

Lifelines COVID-19 Cohort authors are H. Marike Boezen†[1], Jochen O. Mierau[2,3,9], H. Lude Franke[4], Jackie Dekens[4,6], Patrick Deelen[4], Pauline Lanting[4], Judith M. Vonk[1], Ilja Nolte[1], Anil P.S. Ori[4,5], Annique Claringbould[4], Floranne Boulogne[4], Marjolein X.L. Dijkema[4], Henry H. Wiersma[4], Robert Warmerdam[4], Soesma A. Jankipersadsing[4], Irene van Blokland[4,7], Geertruida H. de Bock[1], Judith GM Rosmalen[5,8], Cisca Wijmenga[4]. Correspondence with lead author Lude Franke: l.h.franke@umcg.nl

[1] Department of Epidemiology, University of Groningen, University Medical Center Groningen, Groningen, The Netherlands

[2] Department of Economics, Econometrics & Finance, Faculty of Economics and Business, University of Groningen, Groningen, The Netherlands

[3] Lifelines Cohort Study and Biobank, Groningen, The Netherlands

[4] Department of Genetics, University of Groningen, University Medical Center Groningen, Groningen, The Netherlands

[5] Department of Psychiatry, University of Groningen, University Medical Center Groningen, Groningen, The Netherlands

[6] Center of Development and Innovation, University of Groningen, University Medical Center Groningen, Groningen, The Netherlands

[7] Department of Cardiology, University of Groningen, University Medical Center Groningen, Groningen, The Netherlands

[8] Department of Internal Medicine, University of Groningen, University Medical Center Groningen, Groningen, The Netherlands

[9] Team Strategy &External Relations, University of Groningen, University Medical Center Groningen, the Netherlands

## Author Contributions

**Conceptualization:** Mark P. Mobach.

**Data curation:** Mark P. Mobach.

**Formal analysis:** Mark P. Mobach.

**Methodology:** Mark P. Mobach.

**Validation:** Mark P. Mobach.

**Visualization:** Mark P. Mobach.

**Writing – original draft:** Mark P. Mobach.

**Writing – review & editing:** Mark P. Mobach.

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
