## [Decision Letter · Decision Letter 0]

15 Nov 2022

PONE-D-22-27095Workplace impact on employees: a Lifelines Corona Research Initiative upon a return to workPLOS ONE

Dear Dr. Mobach,

Thank you for submitting your manuscript to PLOS ONE. After careful consideration, we feel that it has merit but does not fully meet PLOS ONE’s publication criteria as it currently stands. Therefore, we invite you to submit a revised version of the manuscript that addresses the points raised during the review process.

ACADEMIC EDITOR:Write a structured abstract with justification of the rationale behind the researchThe introduction should be structured also see revieviwer 1 commentA justification of your methodology is needed and a finally a thorough language editing is needed for spellings and grammer etc .Please submit your revised manuscript by Dec 30 2022 11:59PM. If you will need more time than this to complete your revisions, please reply to this message or contact the journal office at plosone@plos.org. Please include the following items when submitting your revised manuscript:A rebuttal letter that responds to each point raised by the academic editor and reviewer(s). You should upload this letter as a separate file labeled 'Response to Reviewers'.A marked-up copy of your manuscript that highlights changes made to the original version. You should upload this as a separate file labeled 'Revised Manuscript with Track Changes'.An unmarked version of your revised paper without tracked changes. You should upload this as a separate file labeled 'Manuscript'.If applicable, we recommend that you deposit your laboratory protocols in protocols.io to enhance the reproducibility of your results. Protocols.io assigns your protocol its own identifier (DOI) so that it can be cited independently in the future. For instructions see: https://journals.plos.org/plosone/s/submission-guidelines#loc-laboratory-protocols. Additionally, PLOS ONE offers an option for publishing peer-reviewed Lab Protocol articles, which describe protocols hosted on protocols.io. Read more information on sharing protocols at https://plos.org/protocols?utm_medium=editorial-email&utm_source=authorletters&utm_campaign=protocols.

We look forward to receiving your revised manuscript.

Kind regards,

Helen Onyeaka, PhD

Academic Editor

PLOS ONE

Journal Requirements:

2.Please provide additional details regarding participant consent. In the ethics statement in the Methods and online submission information, please ensure that you have specified what type you obtained (for instance, written or verbal, and if verbal, how it was documented and witnessed). If your study included minors, state whether you obtained consent from parents or guardians. If the need for consent was waived by the ethics committee, please include this information.

5. One of the noted authors is a group or consortium Lifelines Corona Research Initiative. In addition to naming the author group, please list the individual authors and affiliations within this group in the acknowledgments section of your manuscript. Please also indicate clearly a lead author for this group along with a contact email address.

Reviewers' comments:

Reviewer's Responses to Questions

**Comments to the Author**

1. Is the manuscript technically sound, and do the data support the conclusions?

Reviewer #1: Yes

Reviewer #2: Yes

2. Has the statistical analysis been performed appropriately and rigorously? 

Reviewer #1: Yes

Reviewer #2: Yes

3. Have the authors made all data underlying the findings in their manuscript fully available?

Reviewer #1: No

Reviewer #2: Yes

4. Is the manuscript presented in an intelligible fashion and written in standard English?

Reviewer #1: Yes

Reviewer #2: Yes

5. Review Comments to the Author

Reviewer #1: “Workplace impact on employees: a Lifelines Corona Research Initiative upon a return to work”

This is a very interesting study, it investigates whether employees working from home had less favourable scores for concern and facilities upon return to work, but better scores for work quality and health than non-home workers. The results show that home workers had less favourable scores for concern and on-site facilities upon return to work, but better scores for work quality and health than non-home workers.

The paper employed a Mann-Whitney U, logistic regression, and Wilcoxon matched-pairs .

General Comments:

The paper has some errors regarding tenses, and grammar. Please consider editing the language thoroughly.

Abstract:

The abstract lacks a sentence or two setting the background or the rationale for this study. Why should you carry out the study? Why should the reader be interested in reading the paper?

Introduction:

- The introduction should be restructured to capture the background, statement of the problem, current literature, existing gaps, the aim of the study and peculiar contributions of this study.

- Please also include the last paragraph in the introduction which explains the structure of the paper. As it is difficult to know what to expect in the paper.

Methodology:

- Please provide a thorough justification of the method used. The authors state the methods used, however, they do not provide a justification for the choice of method. Please prove a paragraph of method justification.

Overall, this paper is important and very interesting. However, the paper is not publishable in its current state. The paper needs to be well structured, especially the introduction. A thorough language editing should also be carried out. Subject to making the necessary adjustments I recommend its publication.

PLEASE SEND IN THE DATA SET IN A USER FRIENDLY FORMAT AND THE DO FILE, OR R FILE OR PYTHON FILE. I NEED TO VERIFY THE VALIDITY/TRUTHFULNESS OF THE RESULTS TABULATED IN THE PAPER

Reviewer #2: The manuscript is well written and the topic is relevant in understanding workers preferences between remote and onsite types of works following the pandemic. Also, the author has done well in attempting to show the correlation between workplace concern, workplace facilities, workplace quality, and workers health, as well as how these factors influence the workers.

6. PLOS authors have the option to publish the peer review history of their article (what does this mean?). If published, this will include your full peer review and any attached files.

Reviewer #1: No

Reviewer #2: No

---

## [Author Response · Author response to Decision Letter 0]

2 Dec 2022

Response to Reviewers PONE-D-22-27095

Academic editor (Ae):

Comment Ae1. 

Write a structured abstract with justification of the rationale behind the research

Response Ae1. 

Thank you for this useful suggestion. The abstract was revised accordingly.

Comment Ae2. 

The introduction should be structured also see revieviwer 1 comment

Response Ae2. 

The introduction was restructured, also in alignment with the advice of reviewer 1.

Comment Ae3. 

A justification of your methodology is needed and a finally a thorough language editing is needed for spellings and grammer etc .

Response Ae3. 

A justification of the methodology was added in a separate paragraph, also in alignment with the comments of reviewer 1. Moreover, language editing was carried out.

Comment Ae4. 

Response Ae4. 

The manuscript (MS) now meets the PLOS ONE style requirements.

Comment A5. 

Please provide additional details regarding participant consent. In the ethics statement in the Methods and online submission information, please ensure that you have specified what type you obtained (for instance, written or verbal, and if verbal, how it was documented and witnessed). If your study included minors, state whether you obtained consent from parents or guardians. If the need for consent was waived by the ethics committee, please include this information.

Response Ae5. 

Thank you. Although it was expected that the initial text in the methods section (“Before entering the study, all participants signed the informed consent.”) was clear on this issue, the text was adapted to make it more explicit that written consent was obtained. It was changed into: “Before entering the study, written informed consent was obtained from all participants.” These changes were also made in the text of the online submission information.

Comment Ae6. 

We note that the grant information you provided in the ‘Funding Information’ and ‘Financial Disclosure’ sections do not match. When you resubmit, please ensure that you provide the correct grant numbers for the awards you received for your study in the ‘Funding Information’ section.

 

Response Ae6. 

The ‘Funding Information’ and ‘Financial Disclosure’ sections of the MS are now consistent. Moreover, grant numbers are not applicable.

Comment Ae7. 

In your Data Availability statement, you have not specified where the minimal data set underlying the results described in your manuscript can be found. PLOS defines a study's minimal data set as the underlying data used to reach the conclusions drawn in the manuscript and any additional data required to replicate the reported study findings in their entirety. All PLOS journals require that the minimal data set be made fully available. 

Response Ae7. 

The MS is based on data from the Lifelines Cohort Study, Study OV20_00070. Lifelines adheres to standards for data availability. Due to ethical restrictions imposed by the Lifelines Scientific Board and the Medical Ethical Committee of the University Medical Center Groningen related to protecting patient privacy, the data are not publicly available. The data catalogue of Lifelines is publicly accessible on https://www.lifelines.nl/researcher/data-and-biobank/$6102/$6104. All international researchers can obtain data at the Lifelines research office (research@lifelines.nl), for which a fee is required. The Lifelines system allows access for reproducibility of the study results.

This information is also given in the Data Availability statement. 

Comment Ae8. 

Upon re-submitting your revised manuscript, please upload your study’s minimal underlying data set as either Supporting Information files or to a stable, public repository and include the relevant URLs, DOIs, or accession numbers within your revised cover letter. Important: If there are ethical or legal restrictions to sharing your data publicly, please explain these restrictions in detail. 

Response Ae8. 

Please see the above Response Ae7.

Comment Ae9. 

Response Ae9. 

Thank you.

Comment Ae10. 

One of the noted authors is a group or consortium Lifelines Corona Research Initiative. In addition to naming the author group, please list the individual authors and affiliations within this group in the acknowledgments section of your manuscript. Please also indicate clearly a lead author for this group along with a contact email address.

Response Ae10. 

Consortium authors’ names and affiliations as well as lead author’s name and contact email address were added to the acknowledgments section of the MS.

 

Comment Ae11. 

Response Ae11. 

The initial MS was double-checked and additional references in new text fragments were added to the reference list and mentioned in the rebuttal letter.

Reviewers' comments (Rc):

Comment Rc1. 

Is the manuscript technically sound, and do the data support the conclusions?

Reviewer #1: Yes

Reviewer #2: Yes

Response Rc1. 

Thank you.

Comment Rc2. Has the statistical analysis been performed appropriately and rigorously? 

Reviewer #1: Yes

Reviewer #2: Yes

Response Rc2. 

Thank you.

Comment Rc3. 

Have the authors made all data underlying the findings in their manuscript fully available?

Reviewer #1: No

Reviewer #2: Yes

Response Rc3. 

Please see the above Response Ae7.

Comment Rc4. 

Is the manuscript presented in an intelligible fashion and written in standard English?

Reviewer #1: Yes

Reviewer #2: Yes

Response Rc4. Thank you.

Comment Rc5. 

Do you want your identity to be public for this peer review? For information about this choice, including consent withdrawal, please see our Privacy Policy.

Reviewer #1: No

Reviewer #2: No

Review Comments to the Author

Reviewer #1 (R1): 

Comment R1.1. 

This is a very interesting study, it investigates whether employees working from home had less favourable scores for concern and facilities upon return to work, but better scores for work quality and health than non-home workers. The results show that home workers had less favourable scores for concern and on-site facilities upon return to work, but better scores for work quality and health than non-home workers. The paper employed a Mann-Whitney U, logistic regression, and Wilcoxon matched-pairs .

Response R.1.1 

Thank you for your positive evaluation of the article and for your helpful suggestions for how to improve it further. 

Comment R1.2. 

The paper has some errors regarding tenses, and grammar. Please consider editing the language thoroughly.

Response R.1.2. 

Language editing was carried out.

Abstract:

Comment R1.3. 

The abstract lacks a sentence or two setting the background or the rationale for this study. Why should you carry out the study? Why should the reader be interested in reading the paper?

Response R.1.3. 

Thank you for this advice. As suggested, in the abstract some sentences were added to explain the study’s rationale.

Introduction:

Comment R1.4. 

The introduction should be restructured to capture the background, statement of the problem, current literature, existing gaps, the aim of the study and peculiar contributions of this study.

Response R.1.4. 

Thank you for this valuable advice. The introduction was revised and your suggestions were included.

Comment R1.5. 

Please also include the last paragraph in the introduction which explains the structure of the paper. As it is difficult to know what to expect in the paper.

Response R.1.5. 

In the last paragraph of the introduction, a text fragment was added to explain the structure of the paper.

Methodology:

Comment R1.6. 

Please provide a thorough justification of the method used. The authors state the methods used, however, they do not provide a justification for the choice of method. Please prove a paragraph of method justification.

Response R.1.6. 

A paragraph of method justification was added.

Comment R1.7. 

Overall, this paper is important and very interesting. However, the paper is not publishable in its current state. The paper needs to be well structured, especially the introduction. 

Response R.1.7. 

Thank you. Your above suggestions with respect to abstract, introduction, and method justification have been included in this revision.

Comment R1.8. 

A thorough language editing should also be carried out. 

Response R.1.8. 

Language editing was carried out.

Comment R1.9. 

Subject to making the necessary adjustments I recommend its publication.

Response R.1.9. 

Thank you.

Comment R1.10. 

Please send in the data set in a user friendly format and the DO file, or R file, or Python file. I need to verify the validity/ truthfulness of the results tabulated in the paper.

Response R.1.10. 

The MS is based on data from the Lifelines Cohort Study, Study OV20_00070. Lifelines adheres to standards for data availability. Due to ethical restrictions imposed by the Lifelines Scientific Board and the Medical Ethical Committee of the University Medical Center Groningen related to protecting patient privacy, the data are not publicly available. The data catalogue of Lifelines is publicly accessible on https://www.lifelines.nl/researcher/data-and-biobank/$6102/$6104. All international researchers can obtain data at the Lifelines research office (research@lifelines.nl), for which a fee is required. The Lifelines system allows access for reproducibility of the study results.

Reviewer #2 (R2): 

Comment R2.1. 

The manuscript is well written and the topic is relevant in understanding workers preferences between remote and onsite types of works following the pandemic. Also, the author has done well in attempting to show the correlation between workplace concern, workplace facilities, workplace quality, and workers health, as well as how these factors influence the workers.

Response R.2.1. 

Thank you for your compliments, highly appreciated.

---

## [Editor Report · Decision Letter 1]

19 Dec 2022

Workplace impact on employees: A Lifelines Corona Research Initiative on the return to work

PONE-D-22-27095R1

Dear Dr. Mobach,

We’re pleased to inform you that your manuscript has been judged scientifically suitable for publication and will be formally accepted for publication once it meets all outstanding technical requirements.

Kind regards,

Helen Onyeaka, PhD

Academic Editor

PLOS ONE
---

## [Editor Report · Acceptance letter]

22 Dec 2022

PONE-D-22-27095R1 

Workplace impact on employees: A Lifelines Corona Research Initiative on the return to work 

Dear Dr. Mobach:

I'm pleased to inform you that your manuscript has been deemed suitable for publication in PLOS ONE. Congratulations! Your manuscript is now with our production department. 

Kind regards, 

on behalf of

Dr. Helen Onyeaka 

Academic Editor

PLOS ONE